



# Observations of near-inertial oscillations trapped at inclined front on continental shelf of the northwestern South China Sea

Junyi Li[1,2,3], Min Li[1], Lingling Xie[1]*

[1] Laboratory of Coastal Ocean Variation and Disaster Prediction, College of Ocean and Meteorology, Guangdong Ocean University, Zhanjiang 524088, China

[2] Key Laboratory of Climate, Sources and Environments in Continent Shelf Sea and Deep Ocean, Zhanjiang 524088, China

[3] Key Laboratory of Space Ocean Remote Sensing and Application, MNR, Beijing, 100081, China

*Correspondence to*: L. Xie (xiell@gdou.edu.cn)

**Abstract.** Interactions of near-inertial oscillations (NIOs) with other oceanic processes have been broadly investigated in recent years. This study presents observed NIOs from a seabed-based platform, which are trapped at an inclined front with strong stratification on the continental shelf of the northern South China Sea (NSCS) during January-February 2021. The current observations reveal that four NIO events occurred induced by wind bursting. Under the same wind forcing, NIO currents in the third event increased from <0.05 m s$^{-1}$ to about 0.08 m s$^{-1}$. The mechanism analysis shows that the amplitude of NIOs was modulated by the shoaling depth of the mix-layer induced by the inclined front, and trapped in the upper layer. More important, the energy transferred from front to NIOs provides a new insight into the NIO dynamics. In addition, a remarkable non-linear interaction (*fD1*) between NIOs and diurnal spring tide occurred at the front zone owing to a strong vertical current shear at the depths of 20–50 m. The underlying physical phenomenon of this observation would be important for energy exchanges in the oceans.

## 1. Introduction

Near-inertial oscillations (NIOs) are internal motions with frequencies close to the Coriolis frequency (Alford 2003). They are a common phenomenon in the ocean and are a response to the passage of either winter storm or tropical cyclone (D'Asaro 1985; Li et al. 2021a). A typhoon-induced upper ocean near-inertial current could be as fast as 1.5 m s$^{-1}$ (Zhang et al. 2022a). During this response, NIOs could not only transfer surface wind energy into the deep ocean, but also drive vertical transport of the water (Whalen et al. 2020). Wind-forced NIOs in a homogeneous water column lead to a barotropic current that is phase shifted by 180° from the surface current (Xing; Davies 2005). In a stratified ocean, NIOs are



excited in the mixed layer and transfer energy through the thermocline to the deep region (Chen et al.
      1996). NIOs contribute most of the velocity shear to the mixed layer (Zhang et al. 2021a). Therefore,
      NIOs are important for energy exchanges in the oceans (Hong et al. 2022).

      Interactions of NIOs with other ocean processes have been investigated in previous studies. The
      energy exchange between eddies and wind-forced NIOs is attributed to strain variance based on a slab
mixed layer model (Jing et al. 2017). Chen et al. (2021) found that self-advection could cause growth of
      near-inertial oscillations in the region of anticyclonic forcing. Xu et al. (2022) observed trapped NIOs in
      a propagating anticyclonic eddy. The nonlinear interaction between inertial and tidal currents induces a
      new signal and accelerates damping of mixed layer NIO (Guan et al. 2014; Liu et al. 2014). Moreover,
      the frequency of NIOs would shift to a super-inertial and sub-inertial frequency modulated background
vorticity or current (Chen et al. 2015; Kawaguchi et al. 2020; Kunze 1985).

      In the coastal zone, where the water depth is shallow and stratification is influenced by runoff, the
      amplitude of NIOs is controlled by the depth of the mixed layer and the prevailing stratification conditions
      (D'Asaro 1985; D'Asaro; Perkins 1984). Xing; Davies (2002) found that the nonlinear current induced by
      NIOs and tides reaches a maximum in the shelf edge region. The NIOs trapped by the sloping isotherms
in the frontal region have been demonstrated using a cross-shelf model (Xing; Davies 2005). The
      boundary of the coastal zone would also reflect NIOs and impact the energy propagation (Ivey; Nokes
      1989).

      In addition, interactions between NIOs and the front have been investigated using numerical models.
      Xing; Davies (2004) found that internal waves with a sub-inertial frequency are trapped between the well-
mixed region and the front. These NIOs could be modified by interaction through the nonlinear
      momentum terms in the front zone (Davies; Xing 2005). Interaction between NIOs and the frontal vertical
      circulations could generate wave beams that radiates down from the fronts (Thomas 2019). In nonlinear
      simulations, wave-wave interactions would result in a thicker layer over the front (Grisouard; Thomas
      2015). Grisouard; Thomas (2016) demonstrated that extra energy of NIOs originates from the front. Xing;
Davies (2004) also demonstrated the occurrence of enhanced surface inertial energy associated with the
      front using a numerical model.

      The northern South China Sea (NSCS) is characterized by a wide continental shelf (Figure 1a). The
      runoff from the Pearl River is carried to the shelf by the Guangdong Coastal Current (GDCC) in winter.
      The thermal and salinity fronts are commonly observed on the shelf. The strongest thermal fronts on the
shelf occur in winter (about 50 km from the coastline) in the NSCS, and are approximately aligned with




the 20–100 m isobaths (Jing et al. 2016). The water is colder, lighter, and more well mixed inside the front than that on the offshore side. In addition, the frequent occurrence of winter storms and strong winds would strongly impact the water mixing in winter in the NSCS, which would induce the NIOs (Sun et al. 2019; Zhou et al. 2023). All these conditions provide an excellent experimental field for observing of the

interactions between NIOs and fronts.

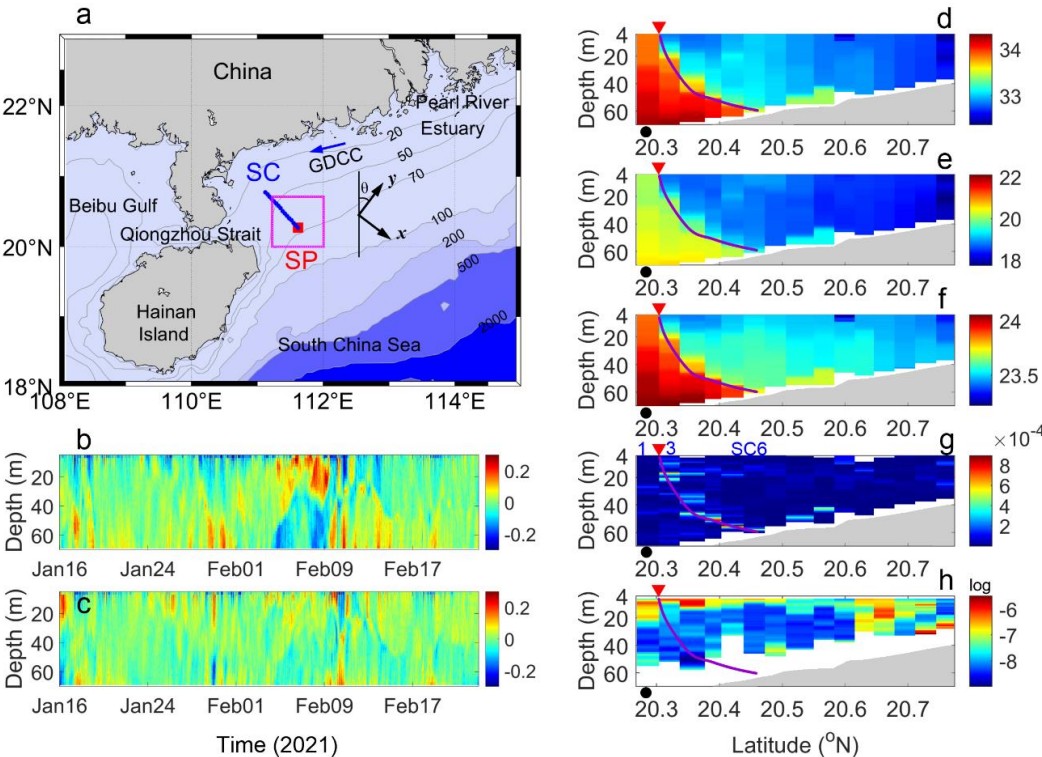

**Figure 1.** (a) Study area. Red square denotes a seabed-based platform station SP. Blue dots represent cruise section SC. Magenta square points out area of wind domain used in this study (0.75°×0.75°). $\theta$

(=36°) is $y$-axis (along-shelf) direction with reference to true north. (b) Baroclinic cross-shelf and (c) along-shelf current components observed at seabed-based platform station SP. Cruise observed (d) temperature, (e) salinity, (f) density, (g) squared buoyancy frequency ($N^2$), and (h) turbulent kinetic energy dissipation rate along section SC. Red triangles on the top denote the front in the sea surface. Black solid dots at the bottom indicate seabed-based platform. Magenta curves represent thermocline along section

SC.



Due to a lack of observation in the NSCS, no efforts have been made to capture the phenomenon of trapped NIOs in the front zone on the continental shelf. In this study, we present the observations of NIOs trapped over an inclined front with a strong stratification on the continental shelf of the NSCS using current data from a seabed-based platform and cruise section measurements. Moreover, the echo intensity obtained from acoustic Doppler current profiler (ADCP) combined with remote sensing data are applied to the partition of the mixed layer. The remainder of this paper is organized as follows. Section 2 presents the data and methods. Section 3 describes the enhanced and trapped NIOs. Section 4 describes an inclined front with stable stratification and analyzes the energy evolution. Section 5 discuss the trapped NIOs and nonlinear waves. Section 6 summarizes the conclusions drawn from this study.

## 2. Data and methods

### 2.1 Data from seabed-based platform

An upward-looking 300 kHz ADCP manufactured by Teledyne RD Instruments (RDI) was deployed over the continental shelf of the NSCS at a bottom depth of 71 m (SP in Figure 1a) to collect current data. The ADCP was configured with a vertical bin size of 2 m and a sampling interval of 10 minutes. The data collection period spanned from January 16 to February 22, 2021. The beam angle of ADCP is 20°. The thickness of blank area is 2 m. After consider the water depth (71 m), the depth of effective observed current data is ranging from 8 to 69 m. Moreover, the data at the depth of 8 m is used carefully.

In addition, the ADCP was equipped with a pressure sensor to measure the water depth $h$. The fluctuations in the sea level, $\eta$, were estimated from $h$. The temporal sampling interval was 10 minutes with an accuracy of about 0.01 m. A three-order bandpass filter was applied to the sea level, and then, a time series of the diurnal tide $\eta_d$ was obtained.

The along-shelf ($u_a$) and cross-shelf ($v_c$) current components were calculated from the northward ($v_n$) and eastward ($u_e$) current components:

$$\begin{pmatrix} v_c \\ u_a \end{pmatrix} = \begin{pmatrix} \cos\theta & -\sin\theta \\ \sin\theta & \cos\theta \end{pmatrix} \begin{pmatrix} u_e \\ v_n \end{pmatrix}. \tag{1}$$

The coordinate system $x$-$o$-$y$ is set as shown in Figure 1a, i.e., the $x$-axis direction perpendicular to the coastline toward the sea is positive and the $y$-axis parallel to the coastline toward the left is positive. $\theta$ (= $\tan^{-1}(U_e/V_n) = 36°$) is the $y$-axis direction with reference to true north. $U_e$ and $V_n$ are the vertical means of $u_e$ and $v_n$.





The barotropic and baroclinic current components were calculated from $u_a$ and $v_c$:

$$u(t) = u_b(t) + u_c(t). \tag{2}$$

where $u_b$ and $u_c$ are the barotropic and baroclinic velocity, respectively, and calculated as

$$u_b(t) = \sum_{i=1}^{m} u_i(t), \tag{3}$$

$$u_c(t) = u(t) - \sum_{i=1}^{m} u_i(t). \tag{4}$$

where $m$ is the bin number in the vertical direction. The barotropic current is the depth-averaged current in this study. The baroclinic along-shelf and cross-shelf currents are shown in Figures 1b–c.

The currents with near-inertial frequencies are extracted from baroclinic current components using a fourth-order Butterworth filter with frequency cutoffs (-3 dB) at 0.0233 h$^{-1}$ and 0.0345 h$^{-1}$ for NIOs, 0.0370 h$^{-1}$ and 0.050 d$^{-1}$ for diurnal baroclinic tide, 0.588 d$^{-1}$ and 0.769 d$^{-1}$ for a summed frequency of the inertial and diurnal frequencies band.

The near-inertial kinetic energy density is estimated via

$$E_f = \frac{1}{2}\rho\left(u_f^2 + v_f^2\right). \tag{5}$$

where $\rho$ (=1024 kg m$^{-3}$) is water density, $u_f$ and $v_f$ are filtered along-shelf and cross-shelf component of near inertial current. Kinetic energy density of diurnal tide could also be calculated by Eq. (5).

## 2.2 Cruise data

The cruise section measurements (blue dots in Figure 1a) were carried out on January 15, 2021. The temperature and salinity profiles (Figure 1d–e) were measured using a Sea-Bird 911plus conductivity-temperature-depth (CTD) sensor. The buoyancy frequency (Figure 1g), $N^2 = -\frac{g}{\rho}\frac{\partial\rho}{\partial z}$, in which $g$ is the gravitational acceleration, and $\rho$ is the water density, was calculated by the temperature and salinity profile. A turbulent velocity shear was measured directly via repeated casts of an MSS90 Microstructure profiler (Sea & Sun Technology). A turbulent kinetic energy dissipation rate ε (Figure 1h) was derived from the turbulent shear wave number spectrum $S_{sh}$ as follows:

$$\varepsilon = 7.5\nu \int_{k1}^{k2} S_{sh}\, dk. \tag{6}$$

where $\nu$ is the kinetic viscosity ($1\times10^{-6}$ m$^2$ s$^{-1}$) of sea water; $k1$ is the lowest wave number (set as 1 m$^{-1}$); and $k2$ is the wave number near the Kolmogorov scale.



**2.3 Sea surface wind and remote sensing data**

The sea surface wind, wind stress and sea surface temperature (SST) data were obtained from the Copernicus Marine Environment Monitoring Service (CMEMS). The temporal resolution of the sea surface wind and stress is 6 h, and the spatial resolution is 0.125° × 0.125°. The data cover the period from January 1 to February 28, 2021. The along-shelf and cross-shelf wind and wind stress components were calculated using the east and north components from Eq. (1). A time series of the area-mean wind

speed was calculated in the magenta square in Figure 1a. The temporal resolution of SST is daily, and the spatial resolution is 0.1° × 0.1°.

The daily ocean color elements (Rrs412, Rrs555, and chlorophyll-a (Chl-a)) were obtained from the moderate resolution imaging spectroradiometer (MODIS) data. The dataset from January to February 2021 is a level-3 product with a spatial resolution of 4 km. The colored dissolved organic matter (CDOM)

and the SST front were calculated using the ocean color elements (Li et al. 2021b; Li et al. 2023c).

**3. Result**

**3.1. Spectral analysis of observed current**

Figures 2a–c show the rotary spectral densities in current records from January 16 to February 22,

2021. The results show that the velocity vector near the $f$ (=0.028 h$^{-1}$) range of 0.8$f$–1.2$f$ mainly rotates clockwise with whole depth, corresponding to a downward propagation of the NIOs in the study area. The frequency shifts of near-inertial peak toward both sub-inertial (0.024 h$^{-1}$) and super-inertial (0.030 h$^{-1}$) frequency bands are also found. The spectral densities are calculated by using a fast Fourier transform with a frequency resolution of 0.001 h$^{-1}$, which could be used to distinguish these frequency bands.

Figure 2d shows the power spectral density (PSD) of the cross-shelf current component within the entire depth range. It is evident that the spectra peaks at the frequencies of the inertial and tidal components ($Di$, $i$=1, 2, 3, 4). The power spectral densities in the inertial period band are more than that in the Diurnal ($D1$) and semi-diurnal ($D2$) tidal frequencies. The result indicates that the energy of NIO is even larger than that of tidal current in the study area. The barotropic tidal current (BT, black curve) is

a primary component. The valleys in the PSD of the barotropic cross-shelf and along-shelf (not shown here) currents suggest that the inertial currents are the baroclinic processes.

In addition, a new spectral peak (denoted as $fD1$) at a summed frequency of inertial and diurnal





frequencies occurs at depths of 24-40 m (red box in Figure 2d).

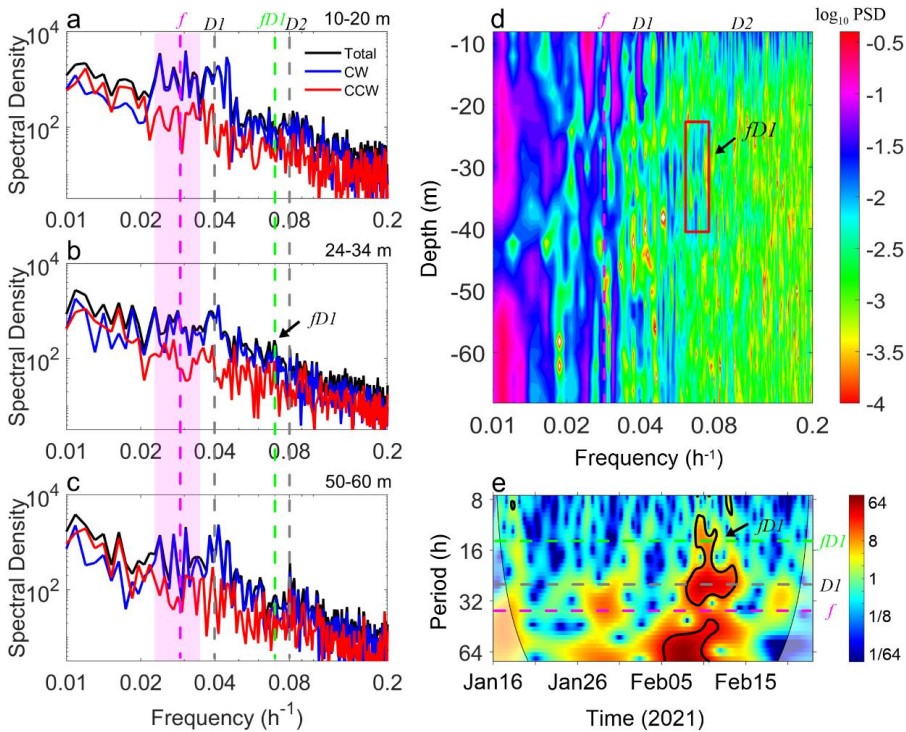

**Figure 2.** Spectral densities of observed current. Spectral densities calculated via velocity averaged in layers of 10–20 (a), 24–34 (b), and 50–60 m (c) at station SP. CW: Clockwise, CCW: Counterclockwise, Total: Clockwise + Counterclockwise. Magenta and green dashed lines represent inertial frequency (*f*) and *fD1*. Gray dashed lines represent diurnal (*D1*), semidiurnal (*D2*) constituents. Magenta shading represent the 0.8*f*–1.2*f* bands. Power spectral density (PSD) of the cross-shelf current component (d). Magenta dashed line represents *f*. The red box shows *fD1* signal in layers of 24–40 m. The frequencies of the inertial (*f*), diurnal (*D1*) and semidiurnal (*D2*) components are denoted in the figure. Wavelet transforms of cross-shelf baroclinic current components at the depth of 24 m (e). The thick black lines are the 5% significance level against red noise. The wavelet power spectrum is normalized.

## 3.2. An enhanced and trapped wind-forced NIO event

To further explore the features of NIO current components in time domain, the currents near-inertial frequencies are extracted from baroclinic current using a fourth-order Butterworth filter (Figures 3b–c). One can see that four prominent NIO events named as E1, E2, E3 and E4, occurred from January 16 to





21, January 27 to 31, February 7 to 14 and February 17 to 21, 2021, respectively. These four events are

corresponding to the four peaks in the sea surface wind as shown in Figure 3a, indicating their wind-

forced nature. The current profiles of NIO show a Mode-1 structure. Nevertheless, the current amplitudes

of these wind-forced NIOs during the four events show a different increase. During the E1, E2 and E4,

the currents of NIOs are about 0.03 m s$^{-1}$. While, the current during E3 is approximately 2 times higher

(~0.08 m s$^{-1}$) with the strongest current above 20 m, although the maximum wind speeds during these

four events are close (~13 m s$^{-1}$). Moreover, the current during E4 was the weakest though the wind speed

is also 13 m s$^{-1}$.

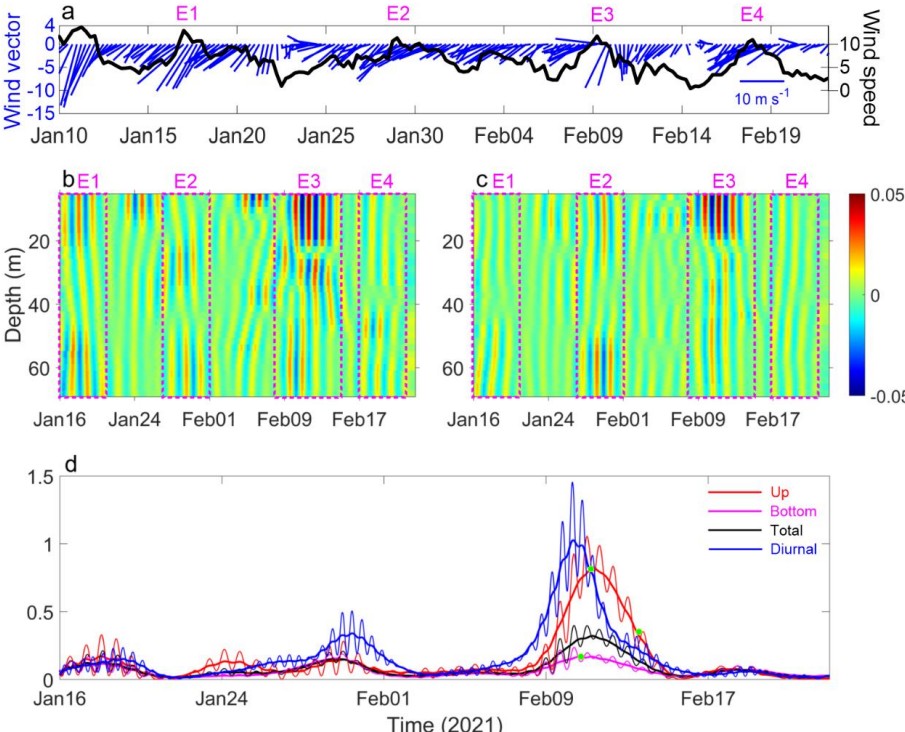

**Figure 3.** Sea surface wind in the study area (a), bandpass cross-shelf and along-shelf current component

of NIOs (b–c), kinetic energy of NIOs and *D1*(d). NIO events in (b–c) are denoted as E1, E2, E3 and E4

(magenta boxes) corresponding to wind speed peaks in (a). In (b–d), a 4-order Butterworth bandpass filter

is with the frequency cutoffs (-3 dB) at 0.0233 h$^{-1}$ and 0.0345 h$^{-1}$ for NIOs, 0.0370 h$^{-1}$ and 0.050 d$^{-1}$ for

diurnal baroclinic tide (blue thin curves). In (d), thick curve is 50 h-smoothed; Up: 8–24 m averaged;

Bottom: 36–68 m averaged; Total: 10–68 m averaged; green dots in red thick curve indicate an *e*-folding

time; green dot in magenta curve points out maximum energy.






Figure 3d shows the evolution of kinetic energy of the NIOs during observation. Firstly, the maximum energy in the upper layer (depth averaged in 10–24 m) is as high as 0.82 J m$^{-3}$ during E3. While, that during E1 was only 0.17 J m$^{-3}$. The kinetic energy of NIOs in the upper layer during E3 is even comparable to that of diurnal tidal current (~1.00 J m$^{-3}$). Secondly, the kinetic energy of NIOs in the

bottom layer (depth averaged in 36-68 m) in these four events are similar, less than 0.18 J m$^{-3}$. Thirdly, the energy in upper layer and total layer are also comparable, less than 0.18 J m$^{-3}$, except during E3.

It looks like that NIOs in E3 are trapped in the upper layer, as the current in the upper layer (< 20 m) is much larger than that in the lower layer (> 20 m). We notice that the wind speeds during these four events are almost the same (~13 m s$^{-1}$). If the stratification characteristics of sea water remain in the same

condition, the current amplitude would be similar in these NIO events. While, the current in E3 was approximately 2 times (~0.08 m s$^{-1}$) larger that during E1, E2 and E4.

These features as shown in Figure 3 imply that the stratification of sea water in E3 is different compared to the other events. Furthermore, in E3, the maximum kinetic energy in the upper layer (the first green point on red curve in Figure 3d) was on February 11, 2021. That in lower layer (green point on

magenta curve in Figure 3d) was on February 10, 2021. The occurrence time of kinetic energy in E3 in the upper layer is lagging that in lower layer. In the next section, we will discuss the impact of the stratification of sea water in these events.

## 4. Inclined front and NIO intensification

As shown by the above results, the current of the wind-forced NIOs during E3 was enhanced and

trapped in the upper layer. The marine environment conditions and potential mechanism of the generation of this enhanced and trapped NIOs during the unique E3 are presented in this section.

### 4.1. Inclined front with stable stratification

No hydrological data is available during NIO events. We assume that the characteristics of the thermal front would be similar during the observation. Therefore, the cruise section measurements carried

out on January 15, 2021 are used to show the characteristics of the thermal front in the study area.

The cruise data revealed that the vertical profile of inclined thermohaline front interface (purple curve in Figure 1f) on the shelf was characterized by extreme stratification with light water above denser



water. The front exhibited a high squared buoyancy frequency ($N^2$) up to $5\times10^{-4}$ s$^{-1}$ (Figure 1g). The turbulent kinetic energy dissipation rate ($\varepsilon$) in the front was exceptionally low, below $4\times10^{-6}$ m$^2$ s$^{-3}$

(Figure 1h). The Richardson number ($Ri = N^2/\left(\frac{\partial u}{\partial z}\right)^2$) was greater than 1. Both parameters indicate that the stratification in this front zone was dynamically stable.

Previous investigations have reported similar characteristics of the front in the north SCS(Zhang; Dong 2021; Zhang et al. 2021b). Jing et al. (2016) identified an upward-tilted front from observations of cross-shelf transects. The low potential vorticity and Richardson number distributed in the coast side zone

of the front indicate a well-mixed condition in the coastal zone. While, the Richardson number in the front was larger than 1 in winter (Jing et al. 2016). These results confirm that inclined fronts frequently occur on the shelf of the NSCS in winter. If the lighter water covers the denser water, the front is in dynamically stable under strong stratification.

## 4.2. Spatiotemporal distribution of thermal front

Figures 4a–e present the spatiotemporal distribution of the thermal front. During events E1, E2, and E4, station SP was situated outside the thermal front in the open sea region (Figures 4b–e). Prior to E3, a notable and continuous decrease in the SST of up to –2°C occurred from February 3 to 7, 2021 (Figure 4a). One can see that station SP was situated inside the thermal front in the coastal region (Figure 4d). In addition, the Chl-a and CDOM concentrations at station SP were relatively high before E4 (Figure 4f),

and they decreased by 50% after February 11.





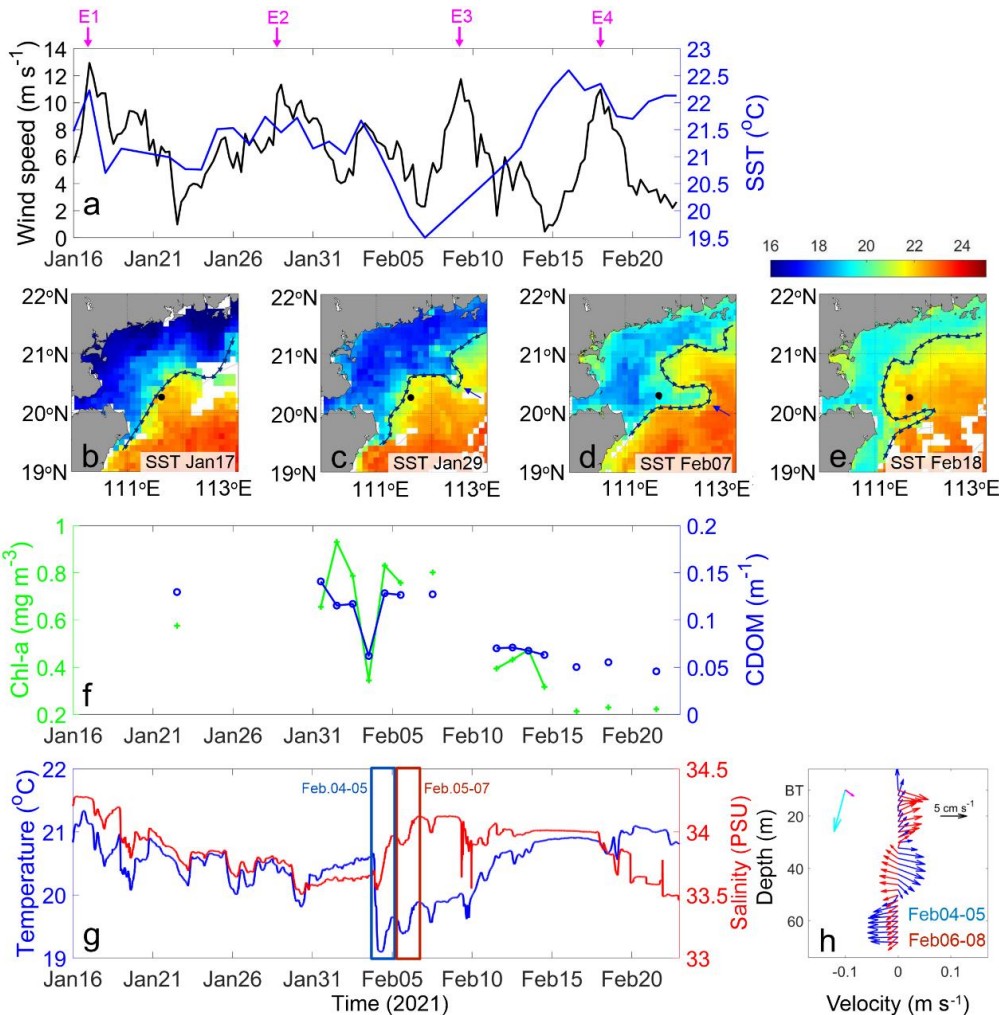

**Figure 4.** Time series of sea surface wind (black) and SST (blue) at station SP (a). E1, E2, E3, and E4 are indicated by the magenta arrows. SST and thermal front (black curve with triangles) in the study area (b–e). Black dot denotes the location of seabed-based platform station SP. Time series of chlorophyll-a (green) and colored dissolved organic matter (blue) observed from MODIS (f). Sea water temperature (blue) and salinity (red) at the sea bottom observed using the seabed-based platform (g). Mean current profiles during February 4–5 and 6–8 (h). BT denotes the barotropic current (blue arrow for February 4-5, magenta arrow for February 6-8).

These satellite observations suggest that the hydrological characteristics of the surface water were similar during E1 and E2 (Figure 4b–c). An intrusion of nearshore surface water due to the front instability



occurred on February 6, 2021 (before E3), which dramatically changed the stratification at station SP (Figure 4d). During E4, station SP was situated outside the thermal front in the open sea region, similar to that during E1 and E2, but it was located further away from the front (Figure 4e).

Moreover, time series of temperature and salinity of sea water (Figure 4g) observed at the sea bottom (about 71 m) provide another perspective. Both the temperature and salinity gradually decreased before February 4, 2021, which have been impacted by the GDCC (Ding et al. 2018). A dramatic increase in the salinity and a decrease in temperature occurred simultaneously on February 5, indicating a coastward transportation. At the bottom, the intrusion of sea water from the ocean sea side was proven by the mean current profile for February 4–5 (blue arrows in Figure 4h). The mean current in the upper layer (>50 m) was seaward, indicating the intrusion of sea water from the nearshore side. This is the reason why the SST at SP decreased after February 3, 2021. After that, the BT current was coastward during February 5–7 (Figure 4h). However, it should be noted that the strong mean current near the sea surface (>20 m) was seaward. The seaward transportation confirms the intrusion of nearshore sea surface water from the SST as shown in Figure 4d.

The depth of the intrusion before E3 is further substantiated and evidenced by the echo intensity observed by the ADCP (Figure 5). The mass concentration of the suspended solids could be estimated using the backscatter intensity measured by the ADCP (Gartner 2004; Lenn et al. 2003). If the suspended solids are uniform in the water column, the echo intensity profile in the vertical water column exhibit an exponential decay curve. One can see that the echo intensity near the sea surface (Figure 5a) increased noticeably from February 5 to 8, 2021. The variations in the mean profile of the echo intensity from February 6 to 7 (Figure 5b) indicate the intrusion of offshore water (>35 m, especially at >20 m) as the nearshore water was characterized by a high turbidity. Subsequently, the inflection points of the echo intensity profile deepened to approximately 50 m from February 15 to 16, 2021. Considering that the front was located far away from station SP during E4, the inflection point of the echo intensity should be caused by the resuspension of sediment.



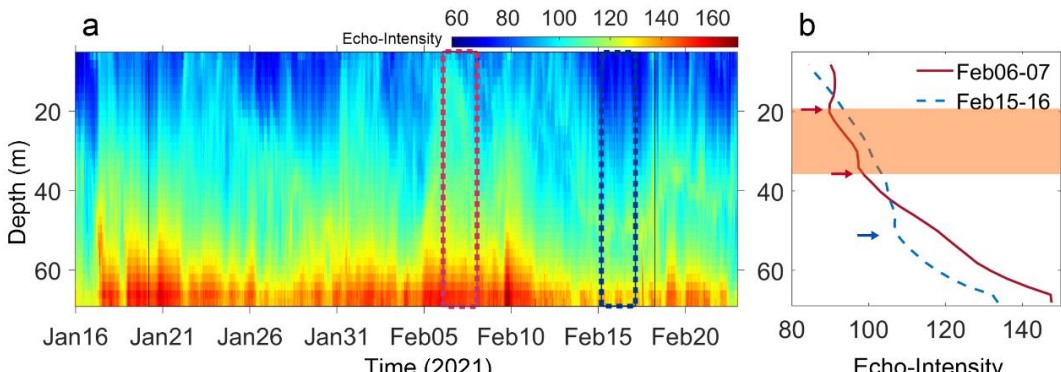

**Figure 5.** Echo intensity observed by ADCP (a) and profile of mean echo intensity (b). Red and blue curves in (b) are the mean profiles during the period framed in (a). Orange shadow marks boundary of different waters. Arrows indicate the inflection points of the echo intensity profiles. ADCP echo intensity could be used to track the compositions and movements of particles in the water.

Overall, the hydrological characteristics were similar during E1 and E2. Before E3, the nearshore water intruded the area of SP in the surface (especially at the depth <20 m), while the water from open sea intruded coastward near the bottom (>35 m). This resulted in an inclined front interface crossing station SP. This sloping front over the seabed-based platform was dynamically stable (see Section 4.1), providing a unique hydrological condition for the generation of NIOs during E3. During E4, the water at station SP should have been almost uniformly mixed above a depth of 50 m (Figure 5b), and it should have had the properties of open sea water (Figure 4e).

### 4.3. Vertical normal mode

An Empirical Orthogonal Function (EOF) is commonly used to decompose the spatial patterns and corresponding time series. Herein we incorporated the EOF analysis with the numerical solution of the normal mode equation.

The vertical profiles of the first EOFs for E1, E2, E3 and E4 (extracted from Figure 3b-c) are shown in Figure 6a. These profiles have a variance contribution of larger than 76.6%. The plus and negative patterns in vertical structures confirm they are Mode-1 NIOs. Moreover, the vertical structure in E1, E2 and E4 are similar. While, that in E3 (blue curve in Figure 6a) changes quickly below the depth of 20 m. The amplitude of Mode-1 NIO below 24 m is about 0.1 m s$^{-1}$, much less than that in upper layer (~0.3 m



$s^{-1}$).

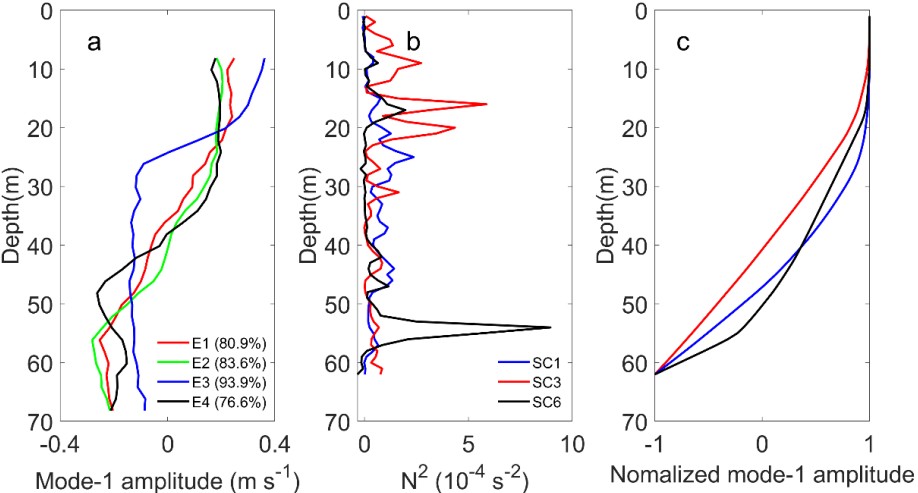

**Figure 6.** Mode-1 vertical structure of currents decomposed by Empirical Orthogonal Function for E1, E2, E3 and E4 (a). Vertical profiles of buoyancy frequency (b) and horizontal velocity (c) corresponding to the Mode-1 of Eq. (8). Blue, red and black curves in (b) are the $N^2$ in SC as shown in Figure 1a, g.

The horizontal NIO current components, *u* and *v*, could be expanded in terms of the vertical normal modes (Chen et al. 2013)

$$(u, v) = \sum_{n=0}^{\infty}(u_n, v_n)(x, y, t)F_n(z). \tag{7}$$

$F_n(z)$ are the eigenfunctions, and the normal modes equation of the internal gravity wave is

$$\frac{\partial}{\partial z}\frac{f^2}{N^2}\frac{\partial F_n}{\partial z} + \frac{1}{r_n^2}F_n = 0, \tag{8}$$

$$\frac{\partial F_n}{\partial z} = 0 \text{ at } z = 0, h. \tag{9}$$

where, $f$ is the Coriolis parameter, $r_n$ is the Rossby radius of deformation of mode *n*.

We use the profiles of buoyancy frequency in SC observed in January 15, 2021 to calculate the horizontal velocity NIO. The red curve in Figure 6b shows a thermal front at the depth of 20 m. The vertical profiles of the first mode derived from the numerical solution of Eqs. (8-9) and the corresponding buoyancy frequency are illustrated in Figure 6b-c. The horizontal velocity (red curve in Figure 6c) corresponding to the NIOs is changing quickly (<20 m) than other curves in Figure 6c. Therefore, the front could modulate the profiles of NIOs obviously.

Therefore, it is credibly that the mixed layer depth is above 20-30 m. Here, we set it as 24 m.



### 4.4. NIOs trapped over front

The above results reveal that a sloping and shoaling front occurred before E3, which was located over station SP. The inclined front induced a two-layer water structure, which was important to the generation of NIOs. During E1, E2, and E4, the depth of the upper layer should be about 50 m. The modified slab model reproduces the NIOs as (Jing et al. 2017; Pollard; Millard 1970)

$$\frac{\partial u}{\partial t} + u \frac{\partial U}{\partial x} + v \frac{\partial U}{\partial y} - fv = \frac{\tau_s^x}{\rho H_u} - ru, \tag{10}$$

$$\frac{\partial v}{\partial t} + u \frac{\partial V}{\partial x} + v \frac{\partial V}{\partial y} + fu = \frac{\tau_s^y}{\rho H_u} - rv, \tag{11}$$

where $(u, v)$ are the cross-shelf and along-shelf velocities in the mixed layer, $(U, V)$ is the geostrophic flow, which is horizontally nondivergent, $H_u$ is the depth of the mixed layer, $\tau_s^x$ and $\tau_s^y$ are the cross-shelf and along-shelf surface wind stress components, $\rho$ is the water density, and $r$ is a damping coefficient. The slab model is the model without the $(U, V)$ as shown in Eqs. (10-11).

In this study, the $e$-folding time is 5 d (green points on the red curve in Figure 3d). Therefore, $r =$ 0.3$f$, corresponding to an $e$-folding time of about 3.3 inertial periods (Guan et al. 2014). The wind stress during observation is shown as Figure 7a. $H_u$ = 50 m during E1, E2, and E4, and $H_u$ = 24 m during E3 based on the results presented in Section 4.2 and 4.3. $(U, V)$ is the background current related to the front (Figure 4d).





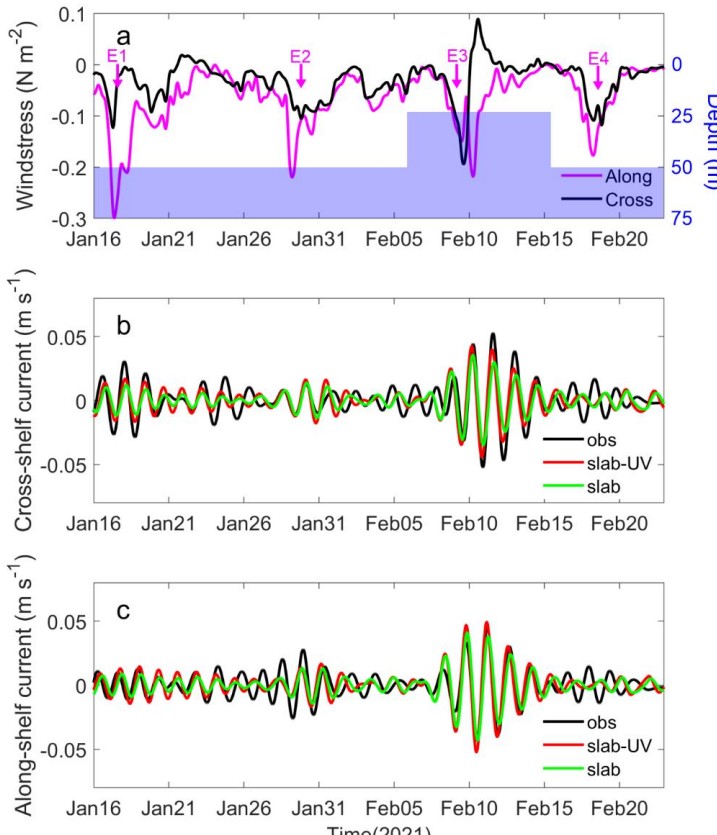

**Figure 7.** Cross-shelf (black curve) and along-shelf (magenta curve) components of wind stress at observation station SP (a). Blue shading in (a) denotes the depth of upper layer. E1, E2, E3 and E4 are indicated by the magenta arrows. Cross-shelf (b) and along-shelf (c) current components obtained using the slab model (red and green curves) and observed (black curve, mean of the current in 8-24 m) NIOs at SP. Green curve is calculated using the slab model without geostrophic flow.

Figures 7b-c show the near-inertial currents calculated using the slab model (green curve) and modified slab model (red curve) compared with the observed near-inertial currents (black curve) in the mixed layer (8-24 m). One can see that the current amplitude of NIOs simulated from the slab model has a good correlation with that of observed current, especially during E1 and E3. During E3, the amplitude of NIO was smaller than that observed NIO. The relative error was less than 30% for the four NIO events. The amplitude of the observed NIO is unexpected higher than that calculated from slab model (green curve) during E3. The simulated and observed results were in antiphase during E4, and the main reason for this is that E3 and E4 were too close.



The water at station SP was well mixed during E1, E2, and E4. The wind momentum rapidly diffused throughout the water column (Xing; Davies 2005). However, the offshore seawater intrusion before E3 created a strong vertical stratification that prevented the wind forcing from penetrating to the deeper layer (Xing; Davies 2005), resulting in a shallower upper layer (mix layer) of 24 m. As a result, the dynamic response to the wind during E3 was double those during the other events with similar wind speeds. Li et al. (2021a) found that NIOs in the stratified area are characterized by large amplitudes in the mixed layer and beneath thermoclines. This result is consistent with the conclusion that stable stratification prevents the wind from penetrating to the deep layer. This indicates that influence of wind during E3 was amplified due to the unique conditions resulting from the intrusion of offshore seawater (Figure 4h).

**4.5. Interaction between front and NIOs**

As shown in Figure 7, the NIO current simulated using the slab model is still smaller than the observed result. Moreover, the maximum amplitude calculated from the slab model occurred one day before than that in the observation. The maximum energy (green points on the red curves) in the up-layer also lagged that in the bottom layer in Figure 3d (green points on the magenta curves). Theoretically, the energy in a wind burst event should inject into up-layer firstly, thus, the energy in the up-layer should lead that in the bottom layer. Therefore, remaining and time-lag energy of the NIO in the up-layer could be induced by either energy transmission or interactions with other processes.

On one hand, the ray-tracing approach can be used to track the transport of a wind-induced near-inertial kinetic energy packet (Kunze 1985). Kawaguchi et al. (2020) used ray-tracing to address the mechanism of NIO's trapping and mid-depth amplification associated with the eddy-wave interaction. The group velocity of near-inertial waves is about several kilometers per day in the central Sea of Japan (Kawaguchi et al. 2020). Ma et al. (2022) found that the horizontal wavelength of NIOs is about 100 km in the SCS, which means that horizontal phase speed is less than 1.0 m s$^{-1}$. However, in this study, the maximum wind stress during E3 occurred in the Pearl River Estuary (not shown here), 400 km northeast of the observation station, which is much greater than the distance of the signal propagation. Therefore, the maximum energy occurred on February 11 was not due to the energy transmission of NIOs.

On the other hand, NIOs can interact with other processes. The modified slab model in Eqs. (10-11) presents relation between geostrophic flow and NIO. As the front is the characteristic oceanic phenomenon during observation. We assume that the geostrophic flow is controlled by the front. Because there is only one observed station, we set the shear and gradient terms of geostrophic flow as fixed values.





Simply, we also set along-shelf gradient of along-shelf current as 0. And the other parameter is about $O$ $(10^{-6}\ s^{-1})$ (Liu et al. 2018). As we can see, the red curve (#1 in Table 1) in Figures 7b-c accord with observed data better than the green curve (slab model without geostrophic flow).

**Table 1**. Coefficients of front characteristic and their values.

| Number | $\partial U/\partial y$ | $\partial U/\partial x$ | $\partial V/\partial x$ | Color in Figure 8 |
|---|---|---|---|---|
| **1** | 0.000005 | 0.000005 | 0.000005 | Blue curve |
| **2** | 0.000001 | 0. 000007 | 0.000001 | Purple curve |
| **3** | 0.000001 | 0.000001 | 0.000001 | Green curve |
| **4** | 0.000005 | 0.000001 | 0.000005 | Magenta curve |

Figure 8 shows the kinetic energy of NIOs from modified slab model and observation. One can see that the kinetic energy of NIOs from slab model is about half of that in observation. While, if consider the energy transfer, the peak of kinetic energy is larger than that from slab model (red curve). Moreover, by

395 using four sets of parameters in Table 1, we can see that the contribution of gradient of cross-shelf current term is larger than the shear terms. Thus, the energy transfer from front to NIO mainly through the gradient of cross-shelf current term. This should be the reason why the simulated amplitude of NIO is smaller than that of observed result in Figure 7.

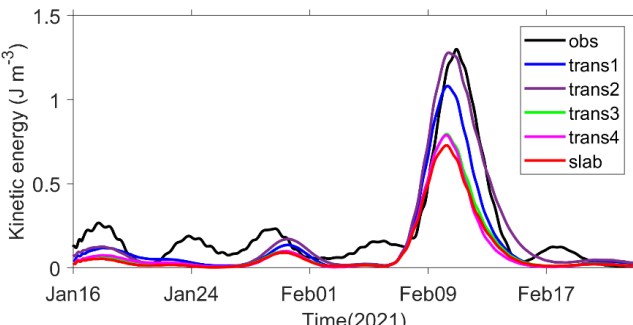

**Figure 8.** Kinetic energy of NIOs from the slab model (red curve), mean observation at the depth of 10–20 m (black curve). Blue, purple, green and magenta curves are that from slab model with the energy transferred from the inclined front (as shown in Table 1).



## 5. Discussion

### 5.1 NIOs in the front zone

Kunze; Sanford (1984) described a wave-mean flow interaction model which predicts trapping and amplification of near-inertial waves in regions of negative vorticity (north hemisphere). In the other area, NIOs are free to propagate out. Li et al. (2021a) presented that the strongest NIOs occur during the restoring period of subtidal flow, indicating an energy transfer from the low-frequency flow to the inertial motion. The average seasonal vorticity in the NSCS (near the station SP) in winter is negative (Zhang et al. 2022b). That should be the reason why the enhanced NIO could be trapped in the study area during E3. During E1, E2, and E4, the observation station was located in the open sea side of the front. Therefore, NIOs would propagate deeper or into the other area freely, resulting a fast attenuation.

Simulations have been investigated to find the energy transfer between front and near-inertial waves. Rocha et al. (2018) found the transfer of energy from balanced flows to existing internal waves. A strong viscous effects introduce additional oscillatory modes that can exchange energy with the front (Grisouard; Thomas 2016). The occurrence with an intensified vertical shear induced by spring tide and sloping front provides a favorable condition for NIOs during E3 in this study. Actually, a large ensemble of wind-generated near inertial waves always extracts energy from the geostrophic flow on average (Whitt 2014). Though the current data from only one observation station was used in this study, the result presents the unique enhanced and trapped NIOs in the front area.

Frequency shift of NIO has been presented in previous studies (Zhang et al. 2021a; Zhang et al. 2022a). The background vorticity induced by other oceanic processes, e.g., eddies and current, can cause a red-shift or blue-shift NIO. The positive (negative) vorticity ($\zeta$) in the ocean shifts the effective inertial frequency ($f_{eff}$) above (below) the local inertial $f$ (Kunze, 1985).Thus, the background vorticity in the NSCS in winter is positive (Li et al. 2023a; Zhang et al. 2022b), which means that the blue-shifted NIOs would occurs. From Figure 2, one can see a slightly blue-shift NIO signal. However, it seems that red-shift NIO signal is the primary component. Li et al. (2023b) found that there are continental shelf waves coincide with NIOs. The sea level variation induced by continental shelf wave would cause the variation of vorticity (Li et al., 2023a). Therefore, the vorticity variation induced by continental shelf waves should be responsible for the red-shift of NIOs in this study.





**5.2 Nonlinear current induced by NIOs and *D1***

*fD1* was observed at the depth of about 24–40 m during E3 (Figure 2). The currents of *fD1* and

diurnal tide were extracted by a fourth-order Butterworth filter (Figure 9). One can see the remarkable

nonlinear process with the frequency of *fD1* occurred during E3. The depth for *fD1* occurred was about

20–50 m, where is the zero-crossing depth for spring tide and NIOs during E3. The amplitude of *fD1*

currents during E3 increased from less 0.02 m s$^{-1}$ to about 0.04 m s$^{-1}$. The current in cross-shelf direction

is larger than that in the along-shelf direction. There seems no *fD1* current occurred in E1, E2, and E4.

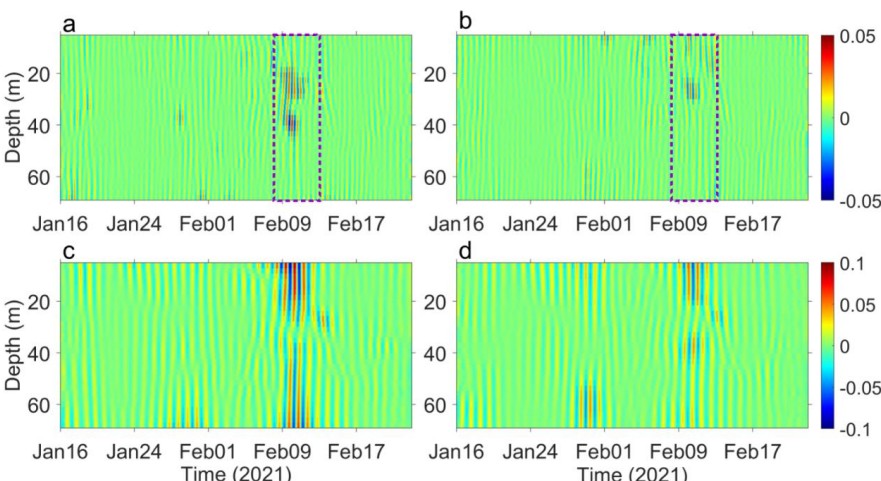

**Figure 9.** Cross-shelf (a) and along-shelf (b) component of *fD1*. Diurnal baroclinic cross-shelf (c) and

along-shelf (d) tidal current. Purple boxes in (a-b) indicate *fD1*. A 4-order Butterworth bandpass filter

with frequency cutoffs (-3 dB) at 0.370 d$^{-1}$ and 0.500 d$^{-1}$ for diurnal baroclinic tide, 0.588 d$^{-1}$ and 0.769

d$^{-1}$ for *fD1* band.

    Figures 9 provides clear indications of the occurrence of the spring tide during E3. A weak spring

tide was observed on January 27, 2021 with the current reaching about 0.05 m s$^{-1}$. In contrast, a strong

diurnal current peaking at 0.11 m s$^{-1}$ occurred on February 11, 2021 (Lunar New Year). The vertical

structure of the spring tide indicated that the mode 1 was a predominant component. The depth of zero-

crossing of mode 1 was about 30 m, especially during E3.

    *fD1* was observed as shown in Figure3, Nonlinear wave-wave interaction is a physical energy

transfer mechanism (Guan et al. 2014). *fD1* can be forced by the coupling of NIOs and *D1* via





$$\frac{\partial u_{fD1}}{\partial t} = -u\frac{\partial u}{\partial x} - v\frac{\partial u}{\partial y} - w\frac{\partial u}{\partial z}, \tag{13}$$

$$\frac{\partial v_{fD1}}{\partial t} = -u\frac{\partial v}{\partial x} - v\frac{\partial v}{\partial y} - w\frac{\partial v}{\partial z}. \tag{14}$$

The non-linear interaction that occurred in the region of the thermocline was associated with current shear in the inertial oscillation ($\frac{\partial u}{\partial z}$) and the vertical velocity ($w$) due to the tide (Xing; Davies 2002). In the front zone, the nonlinear horizontal momentum advection term is as important as vertical shear (Xing; Davies 2005). While, the front was 10 km away from the station SP in E3, which mean the horizontal shear would be much smaller than the vertical shear.

During the Lunar New Year, the spring tide reached its maximum (Figures 10a). If $w = \frac{\partial \eta_d}{\partial t}$, and $\eta_d$ is the sea level of the diurnal tide (Guan et al. 2014), the vertical velocity of diurnal tide during E3 was stronger than that during other events. Moreover, the current shear in the inertial oscillation was strong due to the stable stratification. From Figure 10b, one can see that strong current vertical shear was as high as 0.3 s$^{-1}$ during February 4–14, 2021. The depth decreased from 60 m on February 4 to about 30 m on February 7. After that, the strong wind burst on February 9, and forced the strong NIOs. The non-linear interaction between NIOs and diurnal tide occurred near the inclined front. Xing; Davies (2005) confirmed that maximum nonlinear interaction occurs in the frontal region by using a cross-sectional model. Thus, the strong vertical current shear led occurrence of the nonlinear current (*fD1*) at the front (about 20–40 m). The presence of strong velocity shear was associated with intensified vertical mixing (Pan; Jay 2009). Therefore, the SST at station SP increased rapidly after February 7, though station SP was on the coastal side of the front (Figure 4e).



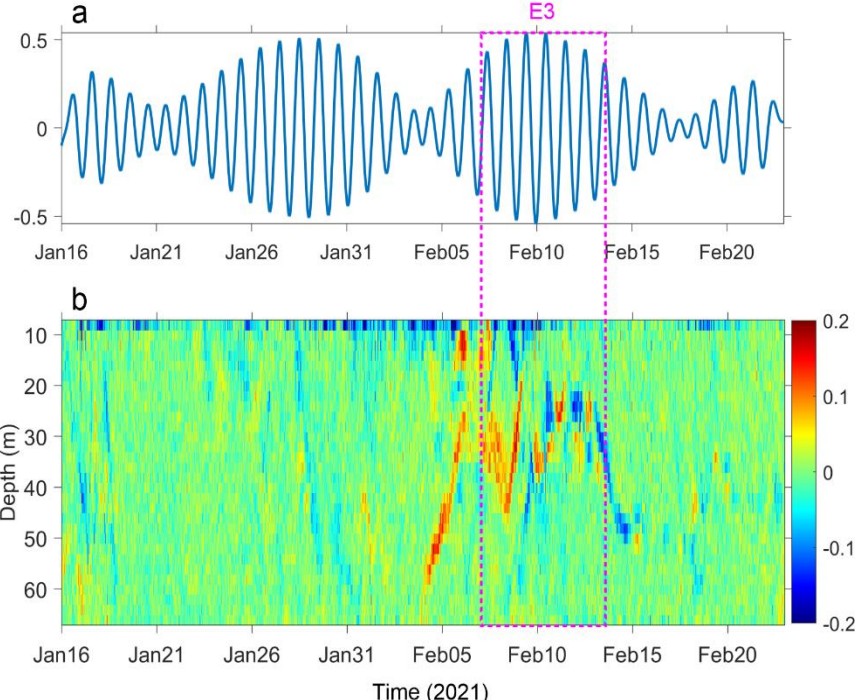


**Figure 10.** (a) Bandpass of sea level of diurnal tide (in m). (b) Vertical shear in cross-shelf direction (in $S^{-1}$).

## 6. Conclusions

In this study, we investigated the characteristics of wind-forced NIOs under strong stratification in the NSCS during January-February 2021. Analysis of the current data from a seabed-based platforms revealed the occurrence of four NIO events, with the NIO current generally below 0.04 m s$^{-1}$ except for event E3. The amplitude of NIO during E3 reached a magnitude of 0.08 m s$^{-1}$, which was twice as high as those during the other events. Moreover, we found that the NIOs were trapped in the upper layer at

depths of less than 24 m. Prior to E3, an inclined front was observed at the observation station, causing changes in the stratification, and facilitating the trapping of NIOs. The observation also provides new insight of energy transferred from front to NIOs. A non-linear interaction (*fD1*) between the NIOs and diurnal spring tide was observed at the front owing to strong current shear in the depth range of 20–50 m in vertical direction. These observations provide evidence of the non-linear interaction between NIOs and

the diurnal spring tide under stable stratification in the coastal ocean. Moreover, the observation of an



unexcepted large NIO during E3 may have been modulated by the energy transferred from the front.

*Data Availability Statement.* Rrs412, Rrs555, Chl-a downloaded from Ocean Color Data Processing System are archived at https://doi.org/10.6084/m9.figshare.23309768.v1. Sea surface wind were

downloaded                                  from                                  CMEMS (https://data.marine.copernicus.eu/product/WIND_GLO_PHY_L4_NRT_012_004).      SST      were downloaded          from          CMEMS          (https://resources.marine.copernicus.eu/product-detail/SST_GLO_SST_L3S_NRT_OBSERVATIONS_010_010/). The shipboard sections data are archived at https://dx.doi.org/10.6084/m9.figshare.19679538.


*Author contributions.* JYL was responsible for writing the original draft. Conceptualization and investigation were handled by JYL, ML and LLX. JYL and ML were responsible for data curation and analysis. LLX acquired funding and provided supervision.

*Competing interests.* The contact author has declared that none of the authors has any competing interests.

*Acknowledgments.* This research was funded by the National Key Research and Development Program of China (2022YFC3104805); National Natural Science Foundation of China (42476027, 42276019, 42176184); Guangdong Basic and Applied Basic Research Foundation (2024A1515012572); Innovation

Team Plan for Universities in Guangdong Province (2023KCXTD015); Guangdong Provincial Observation and Research Station for Tropical Ocean Environment in Western Coastal Water (2024B1212040008).

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
