# Peer review of "Observations of near-inertial oscillations trapped at inclined front on continental shelf of the northwestern South China Sea"

_EGUsphere, 2024_

## Referee Comment (RC2)

**Review of - Observations of near-inertial oscillations trapped at inclined front on continental shelf of the northwestern South China Sea**

**1 General comments**

Intro: Classic references are not shown ndthe objetive andjustificative are not clear

Method: writing requires strong adjustment more explanation of used methods are needed reference of what products aside observations are not provided

Dear Editor,

The study investigates near-inertial oscillations (NIOs) observed from a seabed-based platform on the continental shelf of the northern South China Sea during January–February 2021. Four NIO events were triggered by wind bursts, with the third event showing a current increase from $< 0.05$ m s$^{-1}$ to $\sim 0.08$ m s$^{-1}$ due to modulation by an inclined front. The study reveals that energy was transferred from the front to the NIOs, highlighting a new mechanism for NIO dynamics. Additionally, a non-linear interaction between NIOs and the diurnal spring tide was observed at 20–50 m depth due to strong vertical shear, underscoring the importance of these processes for oceanic energy exchange.

The key points of the paper are as follow:

1. Importance of stratification to the dynamics of NIO.

   - The importance of stratification on the NIO dynamics is well documented. Even studies within the continental shelf being challenging. However, there are nothing new here that deserves to be publish. Beyond, the references used in the paper are deficient and sometimes wrong. The text is definitely not in a good format, which I suggest a strong review.

2. Vertical propagation of energy of trapped NIO

   - The authors claim about vertical energy propagation solely by examining the amplitude of the velocities, which is not enough to arise any conclusion. Moreover, the authors also claim about trapping NIO without disccuss the critical layer which is base of it.

Overall, the study presents some important findings, but it is challenging to follow and understand due to issues with the writing and logical flow. Many statements are not supported by clear evidence or justification, and the reader is often referred to literature that is poorly

or even incorrectly cited. The methods are inadequately described, lacking sufficient justification and proper references. The figures require significant improvements to enhance clarity and better support the study's findings. Additionally, the presentation of the results is weak, and the discussion does not adequately engage with the findings or provide a clear interpretation. A thorough revision would significantly improve the overall quality and coherence of the manuscript.

**I do not recommend that study for publication.**

**Major Points**

1. **Discussion of $\zeta$-refraction:** A large part of the discussion, particularly regarding NIOs and vertical energy propagation, relates to the $\zeta$-refraction process, which is barely discussed and not cited. The authors provide a superficial discussion, implicitly referring to the mechanism without explicitly naming it, but they do not explore it in sufficient depth. Expanding on this mechanism would strengthen the manuscript's scientific foundation.

2. **Trapping of Near-Inertial Waves (NIWs) and Critical Layer:** The trapping of NIWs depends on the presence of a critical layer, which is not shown or analyzed in the paper. Including an analysis of the critical layer would provide valuable insights into the trapping process and improve the overall understanding of the results. Here are a few references that might help the authors analyze and discuss their data more thoroughly:

   - Qu, Lixin, Leif N. Thomas, and Robert D. Hetland. "Near-inertial-wave critical layers over sloping bathymetry." *Journal of Physical Oceanography* 51.6 (2021): 1737–1756.
   - Qu, Lixin, et al. "Mixing Driven by Critical Reflection of Near-Inertial Waves over the Texas–Louisiana Shelf." *Journal of Physical Oceanography* 52.11 (2022): 2891–2906.
   - Thomas, Leif N. "On the modifications of near-inertial waves at fronts: Implications for energy transfer across scales." *Ocean Dynamics* 67.10 (2017): 1335–1350.

3. **Lack of Stability Analysis:** There is no clear stability analysis presented in the manuscript. At times, the authors claim that the front is stable, but they also mention its instability, which creates confusion. A proper stability analysis would clarify this point and make the interpretation more consistent.

4. **Superficial Description of the Turbulent Kinetic Energy (TKE) Dissipation and Richardson Number:** In the methods section, the authors provide a superficial description of the turbulent kinetic energy (TKE) dissipation method and do not

explain how the Richardson number was computed. Did the authors use shear from microstructure data or from the ADCP? What about the stratification—what was the source and sampling rate used? A full and detailed description of the method is necessary. Additionally, the results need to be better explored. Currently, there is only one sentence stating that $\epsilon$ is low at the front and that the Richardson number is higher than 1. If $Ri$ is of order one, submesoscale processes might occur, which are also not discussed. Including a figure showing the Richardson number and buoyancy gradient would significantly enhance the stability analysis and improve the manuscript's overall clarity.

5. **Use of Vague Terms and Lack of Quantification:** Please avoid using subjective terms such as *dramatic* and *strong* without proper comparisons or quantitative evidence. What exactly is meant by *strong stratification* or *dramatic changes*? It is important to provide specific values and comparisons that speak for themselves.

6. **Figures and Captions:** The figures and captions need significant improvement to be self-explanatory. Many figures are missing essential elements such as units, color bars, and axis labels. This makes it difficult to interpret the results without constantly referring to the text. Furthermore, the explanation of the figures is very superficial. Improving the figures and providing clearer captions would make the presentation more effective and accessible.

7. **Data Source Specification:** Please provide a precise and detailed citation for the data source. CMEMS offers a range of different products, so it is essential to specify exactly which product was used to ensure reproducibility and clarity.